# The M-Commerce of Solar Energy Applications: An Analysis of Solar Energy Consumers' Effort Paradox

Cristina Nicolau [1], Ramona Henter [2,*], Mihai Comșiț [3] and Nadinne Roman [4]

1 Centre for Technologies, Inventions & Business S.A. Romania, Faculty of Economic Science and Business Administration, Transylvania University of Brașov, 500036 Brasov, Romania; cristina.nicolau@unitbv.ro
2 Faculty of Psychology and Education Sciences, Transylvania University of Brașov, 500036 Brașov, Romania
3 Centre for Technologies, Inventions & Business S.A. Romania, Faculty of Design and Environment Protection, Transylvania University of Brașov, 500036 Brasov, Romania; comsit@unitbv.ro
4 Faculty of Medicine, Transylvania University of Brașov, 500036 Brasov, Romania; nadinneroman@unitbv.ro
* Correspondence: ramona.henter@unitbv.ro

**Abstract:** The mobile commerce of applications integrating solar thermal collectors, together with their configuring applications, has started to develop. Such applications are not only a business opportunity but also a sustainable and feasible solution for energy consumers who are more and more digitalised. This paper focuses on understanding behaviours in niche markets formed by small-sized and/or isolated consumers who need customized, sustainable and economically efficient applications for heating water for domestic and business use. We chose the focus group interview as the method of research. Primary data were collected in Romania and analysed with Atlas.ti 8. Firstly, the results revealed that consumers' behavioural changes needed for switching to solar energy are influenced by their attitude regarding investment in this market as well as by the perceived social influences and control. Secondly, the results showed the effort paradox of small-sized and/or isolated users of applications integrating solar thermal collectors who shall be considered by developers and sellers alike in providing them with water heating solutions. In terms of business implications, we highlight that the cost-reduction strategy within enterprises is to go green, so major investments in solar technology in order to become energy independent and self-sufficient are envisaged on the long term, whereas the use of digital applications integrating it requires a high level of staff's digital skills and the use of smart devices.

**Keywords:** solar energy; solar thermal collectors; planned behaviour theory; effort paradox; niche markets; m-commerce; consumer behaviour; qualitative research





## 1. Introduction

Out of all the renewable sources of energy, solar energy has become more and more used. Even if large companies seem to be more open to implementing green energy systems to respond to legal and social environmental constraints and reduce costs, small-sized consumers represent a wide target market: small-sized companies (with less than 49 employees) and physical persons (domestic consumers herein referred to as households) would form a noteworthy target.

Solar thermal collectors are sustainable products integrated in applications aimed at heating the water needed for daily activities, domestic or industrial, and/or as a thermic agent for heating. They collect solar energy (which is free of charge) and, by direct conversion, transform it into thermal energy. This process is constant, does not pollute and may be used at low to medium temperatures. Research made in different countries such as Austria, Germany and Denmark shows that the demand for hot water is higher than the supply and that the household necessity of water heating could be supplied by solar energy [1].

For heating water, solar energy is generally used as a complementary type of energy. There are niche markets, however, such as agriculture or tourism, which need it as an

independent source of energy as the other sources are unavailable or too expensive (especially in isolated areas). Our research aimed at identifying and analysing the behaviours of small-sized and/or isolated existing or potential consumers of solar energy technology.

## 2. Literature Review

### 2.1. M-Commerce of Applications Integrating Solar Thermal Collectors

In 2019, mobile telecommunication networks were delivering signals to over 97% of the global population [2]. With such wide spreading, the need for mobile devices such as smartphones and tablets increased, and their use was diversified to match human needs when living and working. Mobile commerce has emerged, however, as a new type of e-commerce. It comprises all the trading activities made via Internet-enabled mobile devices [3,4]. Under such circumstances, m-commerce and the digitalisation of applications integrating solar thermal collectors bring benefits to consumers of green energy, such as having instant access to the latest information, friendly device control through state-of-the-art technology via websites, blogs, webinars, conferences and other promotion and networking events.

Moreover, the m-commerce of green energy applications is influenced by the generations' consuming behaviours. Different generations give different amounts of importance to mobile devices and applications [5] (the most reluctant being Baby Boomers) as well as to green energy and sustainability. Thus, different behaviour patterns are reflected in the inclination towards m-commerce, with Generations Y and Z registering higher rates (of 58% and 49%, respectively, in showing preferences for using mobile devices when buying online [6]). With more and more consumers using mobile applications and demanding do-it-yourself kits, it is high time that the green energy market be marketed online according to consumers' behaviours.

### 2.2. Analysing Green Energy Consumers' Ability to Exert Self-Control: The Planned Behaviour Theory

The consumption of green energy may be predicted by describing the energy users' profile in terms of personal features corresponding to their generations. The planned behaviour theory postulates that consumers' behaviour is determined by their intention (volitional control under certain circumstances) to develop that certain behaviour; intention depends on three factors [7]: (1) the attitude towards the specific behaviour, (2) the subjective norm and (3) the perceived behavioural control, which describes the perception of the ease or difficulty in carrying out that behaviour. The theory is also supported by other studies, as shown in a recent meta-analysis [8].

These three types of factors influence the individual within a specific background created around himself/herself whose individual, social and information characteristics (strengths and weaknesses) accelerate or slow down the development of the intention to perform the desired behaviours (see Figure 1 below). With regard to the green energy consumers, the desire to use green energy and to promote the safety of the environment should come from within the consumers [9], and it may be enhanced by designers', developers', producers' and sellers' mutual work to respond to consumers' needs.

Hence, the consumer's intention to deliberately engage in the behaviour increases once his/her attitude and the subjective norm to the expected behaviour as well become more favourable. At the same time, he/she perceives that more and more that control is needed for that behaviour [11]. In order to increase the possibility that an individual would intend to engage in the desired action and, therefore, increase the chance that the individual would actually complete the action (in this situation, consuming green energy), the offered products/applications integrating solar energy solutions should be able to modify the three predictors: attitude, subjective norm and perceived behavioural control. Consumers' attitudes, subjective norms and beliefs can positively influence his/her intention to purchase green products [12,13]. As environmental concerns become more and more a part of everyday life, consumers generally evaluate green products positively [14], with such a

positive assessment leading to a favourable attitude towards the purchase of green products. Although consumers with high environmental knowledge and abilities tend to pay more for ecological products, the participation of low-eco-skilled people in the production of green products will only slightly increase their adherence to green products [15].

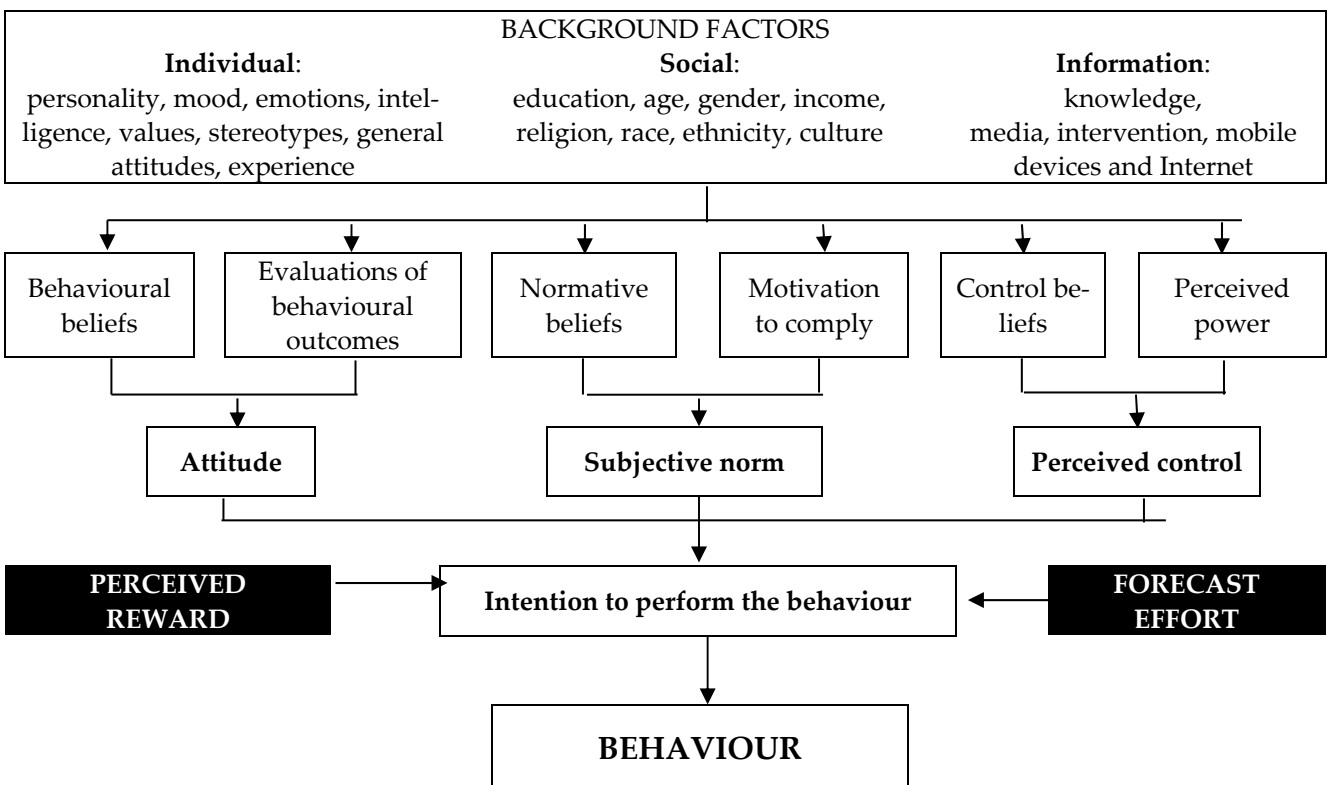

**Figure 1.** The planned behaviour theory adapted from Ajzen and Fishbein [10].

Moreover, consumers' personal norms are influenced by their self-concept and self-image as people who use green products in order to help the environment, their environmental consciousness being an essential construct of green consumer behaviour [16]. The pro-environmental self-identity was found to positively impact green product use [17]. When people believe that their environmentally conscious actions can truly contribute to defending the environment, they will be more motivated to purchase green products [16]. People's beliefs and values regarding the purchase and use of green products were analysed within the theory of consumptions values framework [18], which considers that the consumer's behaviour depends on various consumption values which are independent and have various contributions to the purchase. Previous studies show that consumers who exhibit low levels of environmental concern are unlikely to spend more for green products than the concerned consumers, regardless of how the latter feel about the real impact of their actions [15].

Under such circumstances, solar energy consumers' behaviours are not to be neglected by national policies regulating the green energy markets and/or by businesses developed within this field. Moreover, specific legal regulations should be adopted to support sustainable development through green energy and to protect the consumer itself [19,20]. The end consumers' beliefs when buying renewable energy technology depend on their beliefs about the producers and their products in terms of profitability and efficiency as not all buyers are specialists in green energy; the government protects them by enforcing laws to regulate this market. In addition, the end consumers' perceived control is influenced by such beliefs, and they will tend to buy products from producers that are perceived as being in control of this niche market.

When public services are to be brought closer to the citizen, smart cities respond with their implementation of green energy solutions supported with sustainable policies. Thus, solar energy improves the quality of human life and for such reasons, citizens shall respect the principles of sustainable development. Under such circumstances, digital innovations [21] help public administrations in meeting their purposes and in aligning to the European Community's sustainable energies policies and standards. Therefore, the character of legal relations regarding the management of local public administration should evolve and change positively to suit the conditions in the labour market and significantly innovate [19]. Consumers' intentions to voluntarily use green energy may be highly influenced by the perceived value of the green products/applications, corroborating the effort that the consumer forecasts that he/she will put into the consumption, starting with market research to decide what to purchase and ending with product recycling.

A meta-analytical research study showed that consumers' intention to adopt solar systems is mainly influenced by their perception of benefits; this model, also based on the theory of planned behaviour, revealed that the intention to use solar energy solutions could be predicted by the benefits (defined by general personal motivations, environmental concern, the need of novelty, and subjective norm) and by the perceived behavioural control, which is seen as a main predictor [8].

*2.3. The Effort Paradox of Consumers of Solar Energy Products and Applications Integrating Solar Thermal Collectors*

Effort [22] implies the physical or mental energy needed to do something. It also refers to attempts made which generally involve hard work and determination or to particular situations which demand work to achieve something. Hence, the phenomenology of effort [23–25] shows that the human being perceives it as difficult and tries to avoid it. However, effort has a positive impact on value [26]. When considering applications integrating solar thermal collectors, this impact may be named sustainability, whereas the effort is the price paid by the user. Motivation to behave in a sustainable manner makes consumers of fossil energy prefer products which need more effort in production, distribution and sales, and hence, they replace fossil applications used in their domestic and professional life with green energy, such as solar, wind or wave.

The likelihood to transition to renewable energy is high in terms of declarative willingness, but the actual installation of solar systems also depends on financial issues; hence, both financial (extrinsic) and pro-environmental (intrinsic) motivations should be included in the policies that could foster the transition from fossil fuel use. A monetary incentive (subsidy) for self-consumption of renewable energy was found to play a crucial role in transforming consumers into prosumers, with such a subsidy enabling a change in energy consumption lifestyles at a global level [27].

The solar energy market lacks research on the economic efficiency of the developed products. Users of applications integrating solar thermal collectors may be investigated for the intrinsic properties of their effort itself rather than a property of its product based on the following relationship [26]:

$$\text{Value(effort)} = \text{Reward(effort)} - \text{Cost(effort)}, \tag{1}$$

The reward of economic efficiency in terms of cost reduction and sustainability shall be lower than the prices that consumers of solar thermal collectors pay for products themselves and the repair, maintenance and waste management of the products. This difference between reward and cost balances with the effort itself [28,29]. Hence, this paper analyses the cost, reward and, consequently, the value brought by applications integrating small-scale solar thermal collectors on the energy market and assesses whether the opposing reward and cost create a paradoxical condition in solar energy consumers' behaviours.

Our research is focused on niche markets: small households and businesses which need hot water produced with solar energy only. As a general objective (GO), this piece of research aimed at identifying and analysing the behaviours of small-sized and/or isolated

consumers of applications integrating solar thermal collectors within the framework of the planned behaviour theory [7,11] corroborated to the effort paradox [26]. The specific objectives (SO) of this research are:

SO1. Exploring the attitudes of the consumers on the green energy market, especially of solar energy market;

SO2. Determining the subjective norms generating niche markets and new products and service integrating solar thermal collectors;

SO3. Underlining the perceived control over the decision of using applications integrating solar thermal collectors;

SO4. Exploring the development of m-commerce with solar energy applications.

The connection between the planned behaviour theory and the effort paradox is, in our view, that the former creates the grounds of a behavioural change; hence, consumers' perception of the future rewards that the green energy solutions would bring shall surpass their perception regarding costs of purchase and implementation. For such reasons, this piece of research is of utmost importance as it is centred on the small-scale users for which traditional energy sources are very expensive and/or on the users located in areas which do not have access to other sources of energy and believe that, by heating water with solar thermal collectors, they do reduce environmental degradation [30].

## 3. Materials and Methods

### 3.1. Study Design

Speciality literature [31–34] was analysed before conducting the focus group research, which was focused on obtaining data through open-ended and conversational communication. The main topics are presented in Table 1. Therefore, our research methods allow for in-depth and further probing and questioning of respondents based on their responses, where the interviewer/researcher also tries to understand their motivation and feelings and align to the planned behaviour theory [7,11], corroborating the effort paradox [26]. The theory of planned behaviour is a psychological theory that links beliefs to behaviour. Our research focused on the theory's three core components: attitude, subjective norms and perceived behavioural control, which shape an individual's conduct intentions. The interview started with general discussions aimed at pointing out the importance of solar energy within the energy resources available in Romania and the need to develop and implement solar energy solutions. Then, it continued with advantages and constraints of their use, outlining the prototype matching the needs of small-sized and/or isolated consumers, and in the end, it underlined the development of the m-commerce of the applications integrating solar energy.

**Table 1.** Topics and sub-topics used within the focus group interview.

| Topics | Sub-Topics |
|---|---|
| 1. On the development level of the green energy market | 1.1. Taxonomy of energy resources available to domestic and business consumers <br> 1.2. Pro solar energy at home and at business premises |
| 2. On use of solar thermal collectors in people's lives | 2.1. Current use and limitations of solar thermal collectors <br> 2.2. Technical features of the ideal solar thermal collector <br> 2.3. Developing the m-commerce of solar energy applications |

The use of qualitative methodology created the openness necessary to collect more detailed information [35,36] and find topics that were not considered when the research instrument was designed, such as consumer protection and product regulation, which shall be further investigated. Moreover, the researchers were given the opportunity to use a tool which engages individuals who are difficult to approach with qualitative instruments, as consumers from isolated areas are.

Our sample was formed by using criterion purposeful sampling [37–39], a non-random technique which implies that the researcher first selected a sample with specific criteria and

second collected deep pieces of information with the research instrument (the topic list). Hence, the selection of the right sample for this qualitative research ensured its efficiency and validity [40]. The criteria of sample inclusion were:

(i)　　knowledge of green energy products/applications;
(ii)　　interest in using products/applications with solar thermal collectors at home or at business premises.

We asked participants to self-assess for both criteria on a Likert scale from 1 ('little') to 5 ('strong'), and we included only those people who took an average score of minimum 3.

This qualitative research is of the interpretative phenomenological approach (IPA). The major advantages it offers is that it explored, investigated and interpreted the participants' similar life experiences with no prosecution or distortion [41] by getting to the root cause of the phenomenon [42]. The research technique was the focus group interview, which gave the opportunity to every participant to express, regarding the acquisition (i.e., a solution to produce hot water by integrating a solar thermal collector), its rewards in terms of benefits brought to the consumer and to forecast the effort for acquiring it.

The design of the study implied the use of content analysis [43] so as to explore the primary data which were imported into Atlas.ti 8 and analysed both inductively and iteratively [44] by assigning codes to the qualitative data.

### 3.2. Outcome Measures

As the instrument for collecting primary non-numerical data, this piece of qualitative research used the focus group interview. The research instrument consisted of a list of topics and sub-topics for open discussions, presented in Table 1. The interview started with general discussions aimed at pointing out the importance of solar energy within the energy resources available in Romania and the need to develop and implement solar energy solutions. Then, it continued with advantages and constraints of their use, outlining the prototype matching the needs of small-sized and/or isolated consumers, and in the end, it underlined the development of the m-commerce of the applications integrating solar energy.

### 3.3. Research Population and Participants

Our study is localized in Romania, in the Central Development Region (CDR). This region features continuous innovative development mainly coming from SMEs' capital, firm and entrepreneurial capabilities [45]. The research population is generally represented by Romanian small-sized companies, which were 72,757 in 2022 [46], and households located in urban and rural areas, in total 1,119,779 in 2021 [46] as presented in Table 2. Both categories need to reduce costs with hot water production, and particularly those located in isolated areas need to self-produce the hot water as there is no other energy available.

**Table 2.** Number of small businesses and household in Romanian CDR [46].

| Indicator | 2019 | 2020 | 2021 |
|---|---|---|---|
| Total no. of households in Romania | 9,092,963 | 9,156,311 | 9,587,153 |
| of which, in CDR | 11.65% | 11.67% | 11.68% |
| of which, urban | 59.62% | 59.76% | 60.07% |
| Total no. of enterprises in Romania | 591,259 | 624,206 | n/a |
| of which, in CDR | 11.85% | 11.84% | n/a |
| of which, small companies | 98.13% | 98.37% | n/a |

Our sample consisted of twelve subjects (*n* = 12) who were, first, asked for their informed consent of participation according to the research ethics and data protection law and, second, presented with research scope, objectives and topics of discussion. The research was conducted by two researchers, of which one piloted the instrument and developed the discussions and one was only an observer, assessing both verbal and non-verbal communication. The interviews were audio recorded, which ensured that participants could be identified.

Regarding the sample, the twelve subjects (*n* = 12) were aged between 20 and 65 years old. They were all male and lived and worked in Romanian CDR; 50% of them lived in rural areas and 58.3% worked in rural areas. Their level of education differed from secondary (41.7%) to academic education (58.3%), this independent variable validating their level of knowledge and openness to using solar thermal collectors. Moreover, with regard to their status on the labour market, 66.6% were business owners (4 people ran businesses in service and 4 people, in production), 16.7% were academic teaching and researching staff and 16.7% were a mayor of a village and a volunteer in a local action group.

### 3.4. Data Analysis

Interview audio scripts were transcribed and analysed with the qualitative research software Atlas.ti 8. The pieces of information expressed by participants were coded according to the topics of the focus group interview. We analysed the relationships between the topics referred to by the participants and identified chunks of information within the planned behavioural theory paradigm.

## 4. Results

### 4.1. Changing Solar Energy Consumers' Attitudes

All the participants agreed that the energy had a vital importance in everyday life, stating that they needed it to light and warm the built-in environment (behavioural outcome). For the subjects, the built-in environment mainly meant the buildings in which they lived and/or worked and the necessary infrastructure for performing their daily activities. Irrespective of the individual, social and information factors acting as background factors for behaviour change (i.e., the acquisition of applications integrating solar thermal collectors), participants were concerned (31.58% in relative frequency) about the repercussions that their comfort (behavioural outcome) brought: the development of the built-in environment was irreversibly damaging the environment (behavioural beliefs).

With regard to solar energy, subjects highlighted that this resource is convenient, being given by a 'central shaft'. They identified its main benefits, such as 'light', 'heat', 'radiation' and 'solar gravity' (behavioural outcomes). Therefore, participants expressed a general objective of social responsibility, which was 100% mutual in the sample consisting of the need to exploit as effectively as possible this natural resource so as for people to enjoy it in the built-in environment (behavioural outcome). In this context, solar thermal collectors responding to such needs and applications collecting solar power were perceived as highly sustainable by the participants (behavioural beliefs). This shows that consumers' attitudes for using solar energy are positive and influence their intention to buy and use applications integrating it.

Concerning the types of energy mentioned by the subjects, these are, primarily, the main forms of energy production: fossil energy, nuclear energy and alternative energies such as geo-thermal, wind, solar and tidal power. Secondly, the subjects presented other sources of energy production, namely 'volcanoes', 'forests', 'thunder', 'waves', 'rivers', 'waste' and 'biomasses'.

As consumers of different types of energy, the subjects concluded that, in Romania, the electricity produced from fossil, water, wind and nuclear resources was predominantly used (behavioural beliefs), but they did not seem satisfied with the benefits that these types of power brought them (evaluation of behavioural outcomes). They underlined that the four energy sources previously mentioned were costly on the short and long term (effort). They also emphasized the openness of the population to green sources of producing energy (behavioural belief) but under the conditions that products were sustainable and efficiently presented within producers' and sellers' promotional techniques (effort).

Summing up the responses, the participants expressed their beliefs regarding the use of solar systems, and they presented mainly positive beliefs (in terms of the difference between their positive and negative appreciation of their thoughts on these products and on the outcomes and consequences of their use), which can be considered a basis for

developing their (already) positive attitudes towards solar systems. The participants also offered a positive image on the effort needed to apply these products. Hence, solar systems producers can rely on a favourable attitude towards their product, which could ease the introduction of a larger quantity of their product into the market. Building a favourable attitude towards any product is one of the most difficult marketing tasks. In this case, producers only need to increase this favourable attitude, which can be done by acting on a single aspect that influences attitude: beliefs about the action, consequences of the action and the ease of performing it. Figure 2 shows the determinants of attitude changes, as resulted from the qualitative research herein presented.

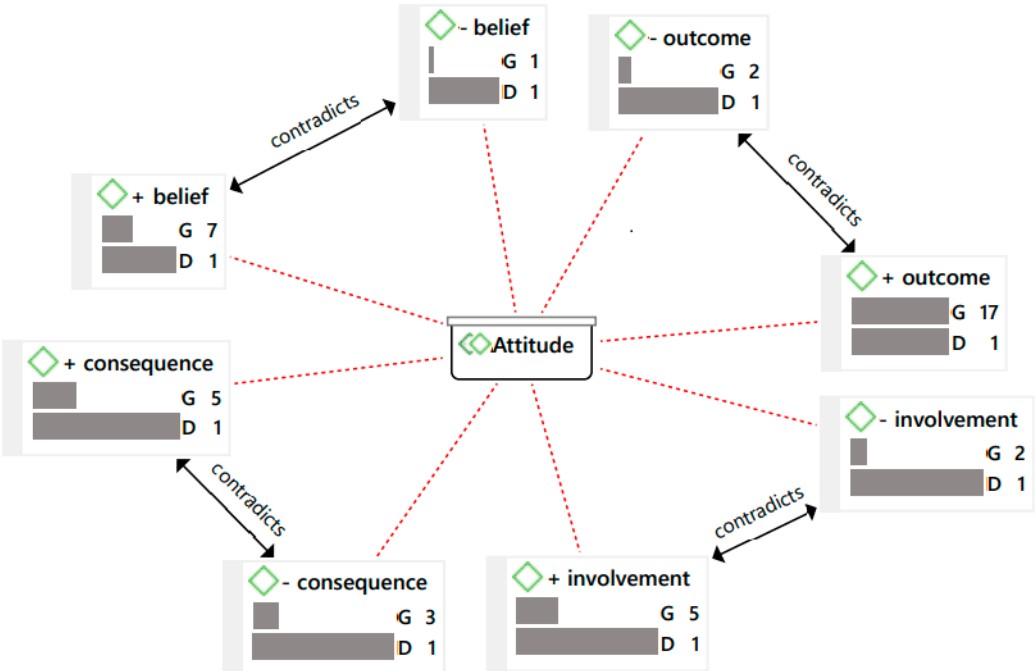

**Figure 2.** Attitude determinants in the participants' view (source: original).

### 4.2. Subjective Norms, Predictors of Solar Energy Consumers' Behaviours

When predicting consumers' intentions to acquire applications integrating solar thermal collectors, subjective norms are not to be ignored by their designers, producers and sellers. Individuals' beliefs about their community's or their relatives' use of solar energy help them perform particular behaviours. In this respect, we identified the main business activity fields that needed to integrate solar thermal collectors, in our participants' views. These fields were assessed as 'able to offer reliability and profitability' both for developers and producers (normative beliefs), as well as for consumers (motivation to comply), on the energy market: construction, agriculture, tourism, service delivery, automotive engineering, wood industry and laser cutting technology. These create important niche markets emerging for the use of small-sized applications integrating solar thermal collectors, and their target consumers are represented by individuals (households) and legal entities (small companies with productive activities or service providers) located in rural or urban areas.

With regard to the range of products and service integrating solar thermal collectors, it shall be divided according to their 'geometric, functional and technological parameters' such as 'collector's surface', 'absorber's active surface', 'distance between tubes', 'operating flow', 'power supply', 'mass', 'area for covering energy requirements' and 'number of panels required'. These parameters shall meet consumers' needs of producing hot water for daily personal or small business activities (motivation to comply), especially in urban and isolated areas which show particularities. Moreover, participants added more qualitative parameters to the discussion, more precisely 'duration of life cycle', 'production cost', 'selling price', 'profitability' and 'installation and maintenance/repair costs', which may

be either rewards or efforts for producers and consumers alike. Hence, participants stated that, by reducing consumers' effort as much as possible, businesses would develop the normative beliefs and increase the motivation to comply.

In order to introduce the use of solar systems as a rule, people need to comply, and compliance is better achieved when the subjective norm underlying it is stronger. Our participants showed motivation to use solar energy, and their beliefs regarding this product are few but positive (as can be seen in Figure 3 below). Their motivation is mostly extrinsic (based on their desire to comply with new rules or new trends and on the pressure they feel in this direction), and producing companies could use these findings to tailor their marketing approach in order to increase the extrinsic motivation which already exists. Consumers' beliefs can be influenced by the outcomes of using the applications integrating solar thermal collectors presented to them in various campaigns or through the example of those who already use them; thus, peer pressure is activated, and it also influences their motivation.

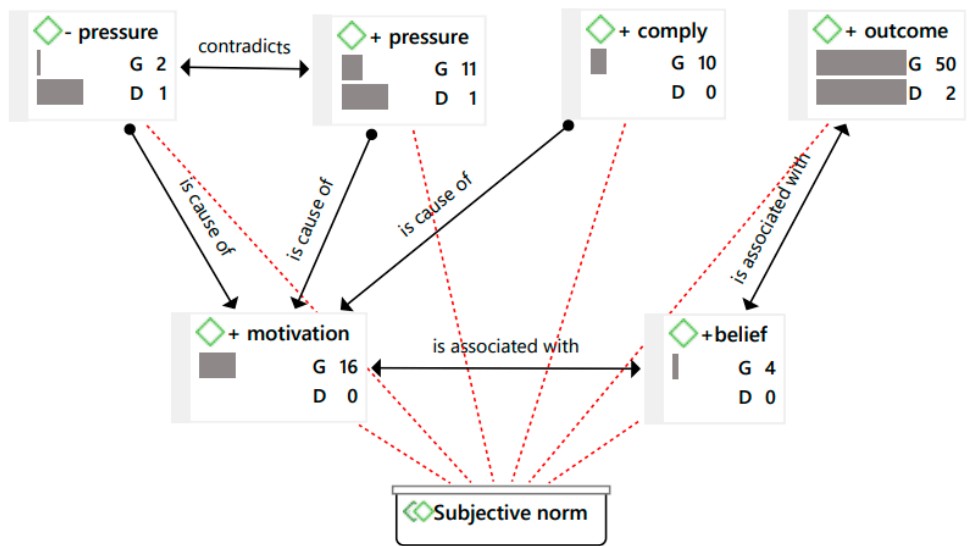

**Figure 3.** The participants' perception on the norms they follow (source: original).

Under such circumstances, niche markets are to be addressed with small-sized applications of solar thermal collectors. As consumers' number is high and their profiles are various, they need applications especially tailored for them; niche markets should not be ignored by designers, developers, producers and sellers. The subjects participating in the study described the 'optimal application integrating solar thermal collectors' meeting their needs (in households and businesses) as a product fulfilling the following requirements that we divided into the two categories [+/− effort] and {+/− reward} in Table 3 below.

These predictors, together with high entrepreneurial vision, shall forecast the future of small-sized applications using solar energy. Further research is needed to analyse whether the ideal solar thermal collectors herein before presented require high investment and generate sufficient revenues for developers and producers while having a selling price affordable for the consumers.

**Table 3.** Predictors of effort and reward in acquiring an application integrating solar thermal collectors.

| Effort | Reward |
|---|---|
| - being cost efficient<br>- being realistic in terms of the coverage degree of the heat requirements<br>- being customized/individualized (at different prices according to users' needs)<br>- being easily accessible (not requiring long times with installation and commissioning)<br>- clearly presenting producers' recommendations on the needed conditions and situations to be used<br>- being tested to avoid unpleasant situations in which the user invests in solar collectors but it does not benefit from their efficiency<br>- being correctly presented to the user (highlighting the advantage of saving energy by heating the thermic agent starting with a certain temperature achieved by using solar collectors) and on the basis of examples of good practice<br>- having heat receivers of other shapes (other than valves/boilers/barrels) | - responding to a small number of needs (excluding large applications which surpass their needs and seem inaccessible)<br>- not generating high maintenance and repair costs<br>- having a lifetime of approximately 5 years (but returning on investment as soon as possible)<br>- being non-piped and unglazed<br>- ensuring the user's convenience<br>- providing a concrete and simple solution for storing heat<br>- being incorporable to construction/building walls (even in the internal walls and/or separation walls of the buildings)<br>- being available in ranges with different physical properties (colour, shape)<br>- using thermic agent other than the ones existing in the market; for example, steam or glycol water have insulated heat receivers so as not to damage soil |

### 4.3. Solar Energy Consumers' Perceived Control of Making the Right Choices

Making the right choice of replacing or supplementing fossil energy with solar energy may be an easy or difficult process, based, on one hand, on consumers' previous experiences and abilities to anticipate constraints, and on the other hand, on the marketing activities undertaken by producers and sellers of applications integrating solar thermal collectors. The analysis of the participants' experiences (including the information provided as a result of other people's experiences shared with them) underlined consumers' need to control the acquisition and use of such applications by the following means:

- spending little time with analysing producers' or sellers' marketing mix: the products with commissioning and maintenance, their prices and promotion, their placement (how close they are to the products so as to see examples of good practice);
- paying for customer-tailored products, matching their real needs and complying in terms of their specifications with what was presented and promoted to them;
- receiving governmental support in terms of reducing legal restrictions for commissioning and recycling and giving funding through national and regional operational programmes;
- benefitting from good practice examples and the expertise of researchers in the field of green energy.

As presented in Figure 4 below, people tend to invest their energy in actions where they feel they are in control, meaning that they are influenced by their perception about their ability to perform that action, the others' interference with it and also by the perceived difficulty of the task. Our participants considered that it is in their power to invest in applications integrating solar thermal collectors, but this was not easy to achieve, probably because of the novelty of the product and their lack of information about it. However, what producers and sellers could use is the examples of efficient and beneficial applications.

Under such circumstances, consumers need control as they put effort in using solar applications, and the reward received from such a use shall be higher than the effort. Cost was considered the highest effort (by all samples), and our participants analysed it per activity within the life cycle of the applications integrating solar thermal collectors: consultancy for needs' assessment, state-of-the-art/innovative product design and development, testing prototypes, customization of the product to respond to needs, increasing new tech life cycle, production, maintenance and repair, recycling at the end of the life cycle, training for using it properly, environment protection and digitalisation.

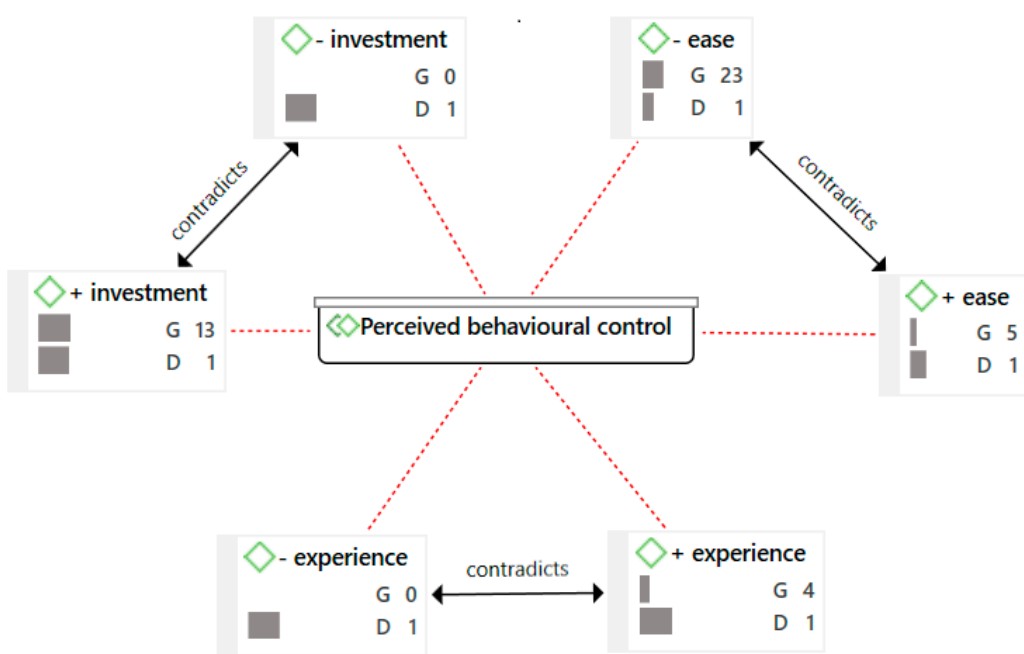

**Figure 4.** Participants perceived behavioural control (source: original).

In contrast, the rewards brought by applications integrating solar thermal collectors and mentioned by the participants were: reducing consumption of natural resources, especially of fossil energy; reducing total costs with energy; the purchase of customer-tailored products give a feeling of uniqueness, whereas consumers respecting the latest norms for green energy have a feeling of satisfaction and pride; fighting against pollution to increase sustainability and maintain health multiplies consumers' self-esteem and belongingness to the community.

*4.4. Digital Applications and M-Commerce*

M-commerce is perceived as a cost [+impediment] in using applications integrating solar thermal collectors. First, participants underlined that language is a barrier for using online apps as apps are not in their mother language. Secondly, subjects acknowledged together that, even if we are in the e-commerce era, the traditional commerce was the most suitable, even if old-fashioned, way of selling energy applications. One participant stated that consumers were familiarized and addicted to traditional commerce, and he pointed out that it would never disappear due to the fact that customers would want to see how such applications work on site and in real time, to analyse the benefits of solar energy and to negotiate offers face-to-face with the producer's or seller's representative. However, m-commerce was recognised to bring the following benefits to consumers of applications integrating solar thermal collectors:

- the geographical borders of markets disappear, and they may be traded worldwide, reducing supply chain costs; with m-commerce, consumers have access to producers and sellers which may provide better and cheaper products, even if they need to pay for shipping;
- design may be done online, asynchronous in an application or synchronous by customer care personnel; hot water project estimators would provide clear information and will promote the applications integrating solar thermal collectors at real parameters, underlining that such applications are available to small-sized consumers, too; thus, the approximate price may be obtained;
- products may be compared and contrasted by technical parameters, brands, prices and even customer ranking (an online marketplace should exist so as to make the

connection among various producers/sellers and international customers, irrespective of their locations);
- online, products and technologies are better explained and tricks and tips are offered, while consumers may participate in live discussions/chats/webinars with experts in the field to learn how to optimize their applications;
- payment and procurement are speeded up;
- transaction costs are reduced and customers may benefit from online financing service;
- installation instructions and self-service support (like in the case of the 'do-it-yourself' products) are provided on the websites of producers/sellers;
- raw materials and spare parts may be available online and easily procured, reducing the system costs due to the short logistics chain.

## 5. Discussions

This innovative study focused on analysing the niche market formed by small-sized users of applications integrating solar thermal collectors in Romania, mainly households and small businesses located in isolated places which have no other energy source. Models of behavioural economics [47,48] consider that effort is inherently costly, whereas effort seekers purely demonstrate that the cost of effort does not exist or it is low [49,50].

The results showed that solar energy consumers' behaviours with regard to their present and future use of integrative applications, both personally and professionally, are more centred on the added value (which takes the form of expectations due to subjects' will to comply with subjective norms) than on the effort put into the acquisition (the high costs of solar energy [51]). However, the use of new technology can be promoted both by financial incentives and by changing people's perception of them. The attitude toward use of specific technologies has a significant positive relationship to price value, therefore is able to influence the behavioural intention to use solar panels; the belief that such technology will be helpful in daily life and that such technology is easy to use as well as the design of an easily accessible technical flow for potential customers can enhance their adoption [52].

Hence, the effort of buying small-scale applications integrating solar thermal collectors is volitional and intentional, as described by Kahneman (1973) [53]. The first three objectives (O1, O2 and O3) of this research were met. With regard to the visibility of users' effort and added value, they positively perceived the association of the two concepts, both in their domestic life as well as in their professional life. Sustainability is, thus, visible through the individual efforts of the users of solar thermal collectors. The studies [54,55] show that the recognition of any effort in self and others provides effort with signalling functions such as dedication, intention or commitment. These are core values of sustainable consumers.

Moreover, the connection between needs, products and technology ("to provide me with energy and fit into the available space") should be considered. The further identification of the impact areas (with immediate needs) and the future customization of any applications of solar thermal collectors for specific groups of consumers shall be considered not only in terms of economic efficiency and economic utility [56] but also focusing on sustainability [57]. Any paradox of efforts in terms of costs and added value shall modify the behaviours of the users of solar thermal collectors. In this regard, the analysis herein offered makes this paper valuable. The consumer behaviour in terms of renewable energy largely depends on the economic incentives, on the possibility to store and later use the energy self-produced (prosumers' motivation is mainly influenced by short-term monetary benefits) or on the amount of the subsidy received at the moment of installing the solar system [27]. Hence, the profitability of the solar energy systems depends on the percentage of self-consumption [27]. In addition, on a higher level, the sustainability of solar systems should be taken into consideration. Renewable energy usage such as solar panels minimizes waste by supporting sustainable development and acknowledging its protective input on the environment; in addition to its large social acceptance, it provides recognition of its benefits [52].

To conclude, this research helps small entrepreneurs in green energy in developing countries deal with the rapid changes and research on solar thermal collectors with the deep analysis of consumers' behaviours in terms of attitude, subjective norms and perceived control, the major drivers of behaviour change. Hence, they are provided with accurate scientific data to help in both the design and development of new applications as well as in writing and implementing a business plan under the circumstances that the future of such products seems sustainability centred. This is why businesses are not to ignore niche markets like rural, isolated and/or very small residential users.

As regards solar energy consumers' attitudes, the results of our research suggest that attitudes may be changed as long as consumers' behavioural beliefs and outcomes are clearly outlined by every individual. From an economic point of view, consumers of applications integrating solar thermal collectors should support the opinion that the natural energy resources are both sustainable (the reward of such products being a healthy and safe environment for the long term) and that they reduce long-term costs (the reward being the reduction of effort of acquisition, installation, maintenance and service).

Under such circumstances, m-commerce shall bring a clear and cohesive organisation of the information that customers need in making their choices of purchase. Currently, remote monitoring applications are available for mobile devices, and their geolocation functions do detect how solar thermal collectors work in real time. Thus, according to our participants, m-commerce represents the added value for the consumers of energy applications, but it depends on the availability and signal quality of the mobile communication networks.

As future research directions, a more comprehensive list of predictors could be used, [8] and researchers could focus on the personal and environmental benefits to better explain consumer behaviour. With the current goal of Europe becoming independent from the use of traditional energy resources, the use of solar systems could be one of the ways to do that. The consumers' personal benefits are rising; therefore, it may be a suitable time to introduce more green energy resources on the market as part of Europe's sustainable development. The current energy crisis brings forth the need to use more renewable energy resources, which are both less expensive than fossil fuels and more environmentally friendly. Solar systems can be remotely controlled and their energy stored or offered to other users. Even if solar energy cannot be used throughout the year at the same peak performance, not only individuals can benefit from the solar systems but also the entire ecological system. Consumers' behaviour is dependent on several factors, including the attitude towards the product. The promotion of solar systems use should focus not only on offering monetary incentive (subsidy) for self-consumption but also on the social perception of the use of sustainable energy. In current times, environment protection has been gaining a similar importance to financial incentives.

## 6. Conclusions

The present study was conducted at the intersection of behavioural sciences, ecology and electronics, as the environmental challenges can be dealt with only through teamwork. Finding a common ground for the technical requirements and the typical green energy consumers' psychological profile is our contribution, which offers the guiding lines in producing and promoting solar systems and offers the manufacturers of integrated solar collectors and integrated mobile applications valuable information on the effective distribution of efforts for the production of products/applications for the niche market of households and small businesses. The major social aspect underlined was the consumer behaviour which should never be neglected by product designers and developers in the solar energy market.

This paper identified respondents' behaviours with regard to solar energy. It spotted new potential users, new niche markets and new areas to design, develop and implement small-sized applications incorporating solar thermal collectors in urban and rural environments in developing countries. Much more detailed quantitative research using a tool

with greater accuracy should be used for larger and more representative samples of the population. Such research shall identify, in much more detail, the needs and consumption of small businesses and households located in rural, urban and isolated places in terms of hot water. Our study focused on people's perception and use of solar thermal collectors, but there are other green energy sources not taken into account in the present study, this being one of its limitations. Another limitation of the present study is the lack of female subjects. In addition, another focus group could be conducted with people who had been using thermal collectors for a couple of years so that the veteran consumers could identify the end-users' benefits and constraints regarding the adoption of renewable energy technology. Not studying in depth the technology used by the consumers can also be a limitation of this research paper. Although, at the individual level, the participants declared the support of renewable energy technology in the form of solar systems, an investigation of the available technology on the Romanian market could offer a better understating of consumers' perceptions and also indicate the level of consumers' acknowledgement regarding the possibility and the need to use green energy.

Further studies should include information about the amount and types of subsidies offered by the government to promote the use of renewable energy sources, if any, and the consumers' perception of them. At the same time, different age groups should be investigated as younger people could have a more favourable perception of green energy, and marketing campaigns should be directed to specific target groups.

**Author Contributions:** Conceptualization, C.N.; methodology, C.N.; software, R.H. and N.R.; validation, N.R. and M.C.; formal analysis, C.N. and R.H.; data curation, C.N.; writing—original draft preparation, C.N. and M.C.; writing—review and editing, C.N., R.H. and N.R.; visualization, R.H. and M.C.; supervision, C.N.; project administration, C.N.; funding acquisition, C.N. and R.H. All authors have read and agreed to the published version of the manuscript.

**Funding:** This piece of research was partially undertaken with ERANET co-funding, UEFISCDI grant number 20/2016-06-24, which also supported C.N. presenting a part of the results at the Conference of Innovative Applied Energy (IAPE) 2019, Oxford, UK. The APC was partially funded by Transilvania University of Brașov, Romania.

**Institutional Review Board Statement:** The research was approved by the Project Ethics Commission on 21 November 2016.

**Informed Consent Statement:** Informed consent was obtained from all subjects involved in the study.

**Data Availability Statement:** Not applicable.

**Acknowledgments:** The authors are grateful to the participants in the research and to Ioan Țoțu for his support and expertise in the field.

**Conflicts of Interest:** The authors declare no conflict of interest. The funders had no role in the design of the study, in the data collection, analyses, or interpretation, in the writing of the manuscript, or in the decision to publish the results.

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
