# Peer review of "The M-Commerce of Solar Energy Applications: An Analysis of Solar Energy Consumers’ Effort Paradox"

_electronics, doi:10.3390/electronics11152357_

Round 1

Reviewer 1 Report

The reviewed scientific article is topically up-to-date and important not only for Romanian consumers. However, there is at least a hint of the legislative side of the problem absent, that is, any mention of consumer protection for unfair practices, especially of various intermediaries, as well as the type of contract by which such green energy will be purchased. This issue is addressed by several foreign authors.

In the context of bringing public services closer to the citizen also within the concept of smart cities, and to increase the citizens' quality of life, even under the pressure of circumstances such as the Covid-19 pandemic brought, also while respecting the principles of sustainable development and, to improve the efficiency and quality of the public services provided, many cities and municipalities strive, for example, bring various digital innovations as an option digital payments via QR codes. It is, therefore, necessary in this research problem, to take into account research such as Vladimíra Žofčinová, Čajková, Andrea and Král Rastislav. " Local Leader and the Labour Law Position in the Context of the Smart City Concept through the Optics of the EU" TalTech Journal of European Studies, vol.11, no.2, 2021, pp.3-26. doi: 10.2478/bjes-2022-0001

Due to the complexity of solving the problem, I recommend the authors to also focus on the issue of not only consumer protection but also the safe production of such energy or subsequent protection of the population due to the effects of extraordinary circumstances from the accident. The research work of the authors Srebalová, M., & Filip, S. (2022) Slovak Self-governments` Legislative Aspects of the Possibilities in Dealing with Nuclear and Other Extraordinary Events is suitable. LEX LOCALIS – Journal of Local Self-Government, 20 (3), pp. 545 - 563, doi: 10.4335/20.3.545-563(2022). Since there may be problems with the availability of the work due to a server failure, I am attaching it in PDF.

In the conclusion, which is incorrectly marked with serial number 4, it should be 5, it is necessary to devote at least one paragraph to the perspectives of the authors' further research in this area.

Author Response

Dear Reviewer 1,

We would like to thank you for reviewing our paper and making obvious the need to focus on the legal aspects coming with new products and regulating new markets. We considered your comments and recommendations so as we performed the changes suggested: we  introduced text on legislative aspects and safe production (lines 128-143) and and further research perspectives (lines 549-564 & 581-583)

Moreover, we uploaded the revised manuscript for you to check.

Yours faithfully,

The authors  

Reviewer 2 Report

The topic of the article is relevant, especially in the current energy crisis. The increase in the cost of electricity and heat energy leads to the need to search for alternative energy sources, including solar collectors. Mobile applications in this case are a conduit that provides access to the market of alternative energy sources for various consumer groups and their requirements. The presented study is carried out in the field of studying the behavior of potential and real consumers of alternative energy sources in the acquisition and use of mobile applications that integrate solar thermal collectors along with applications for their configuration.

The strength of the article and its novelty is that the study was conducted at the intersection of behavioral sciences, ecology and electronics. And also the results of the study contain specific requirements for the technical and consumer characteristics of the product, the main recommendations for positioning and promoting the product / applications identified in the process of conducting the study. These research results are undoubtedly valuable information for manufacturers of integrated solar collectors and integrated mobile applications, which will allow the most effective distribution of efforts for the production of products / applications for the market niche of households and small businesses.

Links are relevant.

The tables and figures clearly show the results of the study.

The following points should be noted as weaknesses:

- remove the gap in figure 1 on pages 2 and 3;

- section “2.3. Research population and participants” describes the sample of the study, which consisted of 12 people. To understand the share of the study sample in the general population, I consider it necessary to provide data on the number of households and small businesses in the study region, which may be potential buyers of integrated solar collectors;

- in section “4. Conclusions” more detailed conclusions are needed, showing the specific results achieved, as well as shortcomings and / or uncovered related areas regarding the study.

Author Response

Dear Reviewer 2,

We thank you for your review and appreciations. We have undertaken the changes according to your suggestions and recommendations, presented as follows:

  1. We put Figure 1 on the same page.
  2. We introduced descriptive statistics at lines 271.
  3. We highlighted shortcomings, uncovered areas and specific results, please see lines 495-501, 518-526, 549-564 & 566-575.

Moreover, we uploaded the revised manuscript for you to check.

We look forward to our revised paper being accepted for publication.

Yours faithfully,

The Authors

Reviewer 3 Report

1. After reading Section 1, I can understand the research objectives of this article. However, I cannot find sufficient reasons or references to find valuable information to link to the research objectives. There are so many articles focus on Theory of Reasoned Action using questionnaire collection from consumers in statistical analysis. Please state clearly why you choose qualitative analysis . 

2. Section 2 should be correct as literature reviews.

3. Section 3 should be correct as material and methods. In addition, please state clearly for the  members in the group discussion and the design of the topics and sub-topics. I suppose that I cannot discuss the unbiased results to the qualitative analysis, but I need to know the background of the members in the group discussion. I think they should be experts and/or opinion leaderships. Besides, more discussion for the topics and sub-topics shown as in Table 1.

4. To be a qualitative research article, Section 4 is too short for any contribution. Most of TRA research articles  can also obtain the same results.

5. The conclusion is  numbered the same as discussion, which should be a mistake. In addition, There is no significant contribution  in this section.

Author Response

Dear Reviewer 3,

Thank you very much for having reviewed our article. We have made changes according to your suggestions, as follows:

  1. We stated the reasons for having undertaken the qualitative research methodology at lines 224-229
  2. We split the former introduction into 2 sections: Introduction and Literature review. Please check line 53.
  3. We highlight that we did not use a panel of experts in solar energy as research technique so we were not interested in their background. Please check lines 230-240 where we explained the construction of the sample and the inclusion criteria. Actually, the ideal participant was the person who had average knowledge of the green energy market as well as was interested to use products of this market.
  4. We added text highlighting contributions, please see lines 566-575. Moreover, all tables and figures add value to our paper.

Moreover, we uploaded the revised manuscript for you to check.

We look forward to our revised paper being accepted by you.

Yours faithfully,

The Authors

Reviewer 4 Report

Dear Authors

I find this work really well done. However, I would like to ask you to clarify some points that may not be clear to those who are not familiar with the topic. And to my way of thinking, types of work like these have the ability to be appreciated even by the layman.

1. Abstract is devoid of the main managerial implications 

2. Introduction can be improved literature by providing an overview of social analysis work related to solar https://doi.org/10.1016/j.erss.2021.102339 (perceived benefits) https://doi.org/10.1016/j.erss.2021.102405 (the role of self-consumed energy) https://doi.org/10.3390/su14010280 (impact of technology)

3. The methodology should be explained with respect to the literature, indicating the assumptions made and more details about the experiment conducted. In this way we favor, replicability. It is, however, well structured

4. Compliment results

5. Discussion can be improved by evaluating social aspects.

6. Conclusions are totally absent. I invite you to give light to the main concepts obtained in this work to the methodological and managerial implications and to indicate the main limitations of the work.

Author Response

Dear Reviewer 4,

Thank you for having read and assessed our research article. We highly appreciate your kind words and suggestions. In order to improve the quality of our article, we have responded to your requirements, as follows:

  1. Abstract: we introduced text presenting the main managerial implications, please check lines 24-27.
  2. Introduction: we introduced text according to the references you suggested. To see the references, please see lines 636, 675 & 725
  3. Methodology: we provided more details on the qualitative research undertaken and we included more references (please check lines 224-229) whereas we re-organised the text of sections 3.1 & 3.3.
  4. Discussions: we improved by highlighting the social aspects, please see lines 495-501 & 518-526.
  5. Conclusions: we highlighted the concepts obtained, managerial implications and limitations, check lines 566-575, 581-577 and 580-600.

Moreover, we uploaded the revised manuscript for you to check.

We look forward to our revised paper being accepted for publication.

Yours faithfully,

The Authors

Round 2

Reviewer 1 Report

I am glad that the author/authors accepted my comments and incorporated them into the text, which clearly increased the scientific value of the article as well as the complexity of the research.

Reviewer 3 Report

The  revision is more appropriate than the first version for reading and undersaanding the main contribution of the authors. I recommend to publish this article.

Reviewer 4 Report

All comments are well integrated. Congratulations.